# Analysis of Stress Predictors in *Vaquejada* Horses Running with Different Interval Rest Periods

Clarisse S. Coelho [1,2,3,]*, Ticiane R. P. Sodré [4], Lara N. Sousa [4], Thadeu M. Silva [4], Joana Simões [2,3,5,6], Vinicius R. C. Souza [3], Renata F. Siqueira [7] and Helio C. Manso Filho [8]

1 Mediterranean Institute for Agriculture, Environment and Development, Universidade de Évora, 7006-554 Évora, Portugal
2 Veterinary and Animal Research Centre (CECAV), Faculty of Veterinary Medicine, Lusofona University, 376 Campo Grande, 1749-024 Lisbon, Portugal; joana.simoes@ulusofona.pt
3 Faculty of Veterinary Medicine, Lusofona University, 376 Campo Grande, 1749-024 Lisbon, Portugal; vinicius.souza@ulusofona.pt
4 Escola de Medicina Veterinária e Zootecnia (ESCMEV), Federal University of Bahia (UFBA), Salvador 40170-110, Bahia, Brazil; ticy_sodre@hotmail.com (T.R.P.S.); laranunesmvet@gmail.com (L.N.S.); thadeumsilva@gmail.com (T.M.S.)
5 CIISA-Centre for Interdisciplinary Research in Animal Health, Faculty of Veterinary Medicine, University of Lisbon, 1300-477 Lisbon, Portugal
6 Associate Laboratory for Animal and Veterinary Science (AL4AnimalS), Faculty of Veterinary Medicine, University of Lisbon, 1300-477 Lisbon, Portugal
7 Faculdade de Medicina Veterinária, Universidade Federal de Santa Maria (UFSM), Santa Maria 97105-900, Rio Grande do Sul, Brazil; refarinelli@yahoo.com.br
8 Núcleo de Pesquisa Equina, Universidade Federal Rural de Pernambuco (UFRPE), Recife 51171-900, Pernambuco, Brazil; helio.mansofo@ufrpe.br
* Correspondence: clarisse.coelho@ulusofona.pt; Tel.: +351-935125059

**Abstract:** The purpose of this study was to assess the stress responses associated with *vaquejada* simulating tests (VSTs). Ten clinically healthy $8.9 \pm 4.3$-year-old Quarter horses, weighing $441.3 \pm 25.0$ kg, performed two VSTs five days apart. In the first VST (M1), animals ran three times with a 5-min rest between races; and in the second (M2), they ran with a 15-min rest between races. Horses were examined before (T0) and immediately after the third run (T1) and at 4 h (T2) of recovery. Stress biomarkers included heart rate (HR), heart rate variability (HRV), cortisol, and white blood cell count. All variables were analyzed by ANOVA and Tukey tests, considering $p < 0.05$. A significant reduction in cortisol was observed for pull ($p = 0.0463$) and helper ($p = 0.0349$) horses when they had a 15-min rest between races. The rMSSD and mean R-R values for helper horses were also significantly lower in M2. In conclusion, the 15-min rest interval proved to be better than the 5-min period for both categories of equine athletes used in *vaquejada* mainly for helper horses. A longer rest time between races allowed the organic recovery necessary for these animals to impose a greater applied physical effort load, which is a fact that guarantees good performance and well-being.

**Keywords:** cortisol; heart rate variability; leucocytes

## 1. Introduction

The welfare of athletic horses is a constant concern [1], and physical exercise can be considered a physiological stressor [2–4]. Stressful events lead to a rapid sequence of psychological and biological responses to support increased demands on the body [5–7]. Stress exposure can be adaptive, causing a beneficial modification to body regions, or maladaptive, especially when it cannot be escaped or controlled, leading to learning impairments and psychopathologies [7,8]. Furthermore, when underconditioned or overworked, animals show a depletion of performance capacity, increasing the risk of injuries and potentially decreasing the quality of life [4,9].

Quantifying the exercise-related stress response of horses represents a way to monitor training and welfare [4]. Various biomarkers can be used to evaluate athletic performance [2,10] and, more recently, some have also been as way to characterize welfare, such as cortisol, heart rate and heart rate variability [9,11–13].

Equestrian sport modalities that involve more than one animal species can represent a challenge when ensuring the welfare of all the participants in the practice of the activity. *Vaquejada* represents one of these equine disciplines in which two Quarter horses and their cowboys run alongside a bull on a soft sand track, with one of them (helper horse) guiding the bull and ensuring that it continues running in a straight line, and the other (pull horse) galloping alongside the bull and working together with the other rider to pull the bull down by catching its tail [14–16]. Although involved in the same sport, these two equine athletes perform distinct tasks, and this must be considered in *vaquejada* training and competitions [16]. So, it is crucial to understand the organic adaptations of both athletes when executing this particular exercise.

Considering the hypothesis that there is an ideal rest interval between races in which animals can be worked according to the precepts of equestrian welfare, the main purpose of the present research was to evaluate stress predictors (cortisol level, leukogram, heart rate, and heart rate variability) of Quarter horse athletes submitted to *vaquejada* simulation tests each one with a different rest interval between races (5 min and 15 min).

## 2. Results

Significant effects of the VSTs were observed in the HR of pull and helper horses regardless of the rest interval between races (M1 and M2) with higher values recorded immediately after the races (T1). Also, comparisons between athletes only revealed significant differences in HR in M2 with higher values for pull horses (Table 1). After 30 min of recovery, the registered HR of pull horses in M1 was 47.8 bpm and in M2 was 44.4 bpm, whilst in helper horses, it was 55.2 bpm and 41.8 bpm in M1 and M2, respectively.

**Table 1.** Physiological and blood parameters measured in pull and helper horses after a VST with a 5 (M1) and 15 min (M2) intervals between races.

| | Experimental Period | | | *p* | |
| | At Rest | Immediately after the Three Races | At 4 h of Recovery | Physical Activity | Animal Category |
|---|---|---|---|---|---|
| **Cortisol (μg/dL)** | M1 (5-min interval between races) | | | | |
| Pull horses | $8.78 \pm 1.44$ [ab] | $13.50 \pm 4.93$ [a] | $4.36 \pm 1.19$ [b] | 0.0018 | 0.3872 |
| Helper horses | $8.32 \pm 1.13$ | $13.30 \pm 3.77$ | $9.06 \pm 7.77$ | 0.2780 | |
| | M2 (15-min interval between races) | | | | |
| Pull horses | $7.12 \pm 2.79$ | $7.20 \pm 2.81$ | $4.42 \pm 2.00$ | 0.1906 | 0.1777 |
| Helper horses | $7.82 \pm 2.18$ | $7.86 \pm 1.44$ | $6.62 \pm 2.88$ | 0.6229 | |
| **HR (beats/min)** | M1 (5-min interval between races) | | | | |
| Pull horses | $40.80 \pm 4.60$ [b] | $103.20 \pm 47.87$ [a] | $41.20 \pm 5.59$ [b] | 0.0059 | 0.3107 |
| Helper horses | $45.60 \pm 11.44$ [b] | $95.20 \pm 30.78$ [a] | $51.20 \pm 9.55$ [b] | 0.0005 | |
| | M2 (15-min interval between races) | | | | |
| Pull horses | $43.20 \pm 3.35$ [b] | $135.60 \pm 31.19$ [a] | $44.80 \pm 6.57$ [b] | <0.0001 | 0.0007 |
| Helper horses | $41.00 \pm 5.10$ [b] | $97.60 \pm 19.51$ [a] | $39.60 \pm 6.99$ [b] | <0.0001 | |
| **Total WBC ($\times 10^3$/μL)** | M1 (5-min interval between races) | | | | |
| Pull horses | $8.02 \pm 1.26$ | $9.64 \pm 1.67$ | $9.54 \pm 1.37$ | 0.1810 | 0.5870 |
| Helper horses | $7.92 \pm 1.57$ | $9.84 \pm 2.21$ | $10.62 \pm 3.30$ | 0.2450 | |
| | M2 (15-min interval between races) | | | | |
| Pull horses | $7.68 \pm 1.19$ | $9.60 \pm 2.16$ | $8.46 \pm 1.45$ | 0.2220 | 0.8750 |
| Helper horses | $7.84 \pm 1.82$ | $9.38 \pm 1.37$ | $8.26 \pm 1.02$ | 0.2570 | |

**Table 1.** *Cont.*

| | Experimental Period | | | | *p* |
|---|---|---|---|---|---|
| | **At Rest** | **Immediately after the Three Races** | **At 4 h of Recovery** | **Physical Activity** | **Animal Category** |
| NEU ($\times 10^3/\mu$L) | M1 (5-min interval between races) | | | | |
| Pull horses | $5.78 \pm 0.96$ | $5.66 \pm 1.01$ | $6.66 \pm 0.36$ | 0.1560 | |
| Helper horses | $5.60 \pm 1.01$ | $6.59 \pm 2.00$ | $7.86 \pm 2.73$ | 0.2530 | 0.2560 |
| | M2 (15-min interval between races) | | | | |
| Pull horses | $5.24 \pm 0.84$ | $6.04 \pm 1.08$ | $5.54 \pm 0.81$ | 0.4070 | |
| Helper horses | $5.57 \pm 1.32$ | $6.38 \pm 0.98$ | $5.70 \pm 0.56$ | 0.4230 | 0.4260 |
| LIN ($\times 10^3/\mu$L) | M1 (5-min interval between races) | | | | |
| Pull horses | $2.00 \pm 0.41$ | $3.59 \pm 1.77$ | $2.56 \pm 1.08$ | 0.1550 | |
| Helper horses | $2.00 \pm 0.50$ | $2.85 \pm 0.29$ | $2.49 \pm 0.65$ | 0.0560 | 0.4300 |
| | M2 (15-min interval between races) | | | | |
| Pull horses | $2.17 \pm 0.33$ | $3.32 \pm 1.71$ | $2.60 \pm 0.69$ | 0.2760 | |
| Helper horses | $1.97 \pm 0.60$ | $2.81 \pm 0.48$ | $2.33 \pm 0.43$ | 0.0640 | 0.2830 |

Note: Data expressed as mean values and standard deviation. Different letters in the same line denote significant differences by Tukey test ($p < 0.05$). HR, heart rate; WBC, white blood cell count; NEU, neutrophils count; LIN, lymphocytes count.

A significant difference in the pull horses' cortisol level was observed in M1. No differences were registered for total WBC count, neutrophils and lymphocytes counts (Table 1). However, a strong positive correlation ($p < 0.0001$) was observed between total WBC count and neutrophil count ($r^2 = 0.86$) and between total WBC count and lymphocyte count ($r^2 = 0.70$).

The comparison between mean values recorded for the 5-min interval (M1) and 15-min interval (M2) amid races (Table 2) showed significant differences only for cortisol with lower values recorded in M2.

**Table 2.** Cortisol, total white blood cell count, neutrophils and lymphocytes mean general values according to rest interval time between races (5 min and 15 min).

| Variables | Interval between Races | | *p* |
|---|---|---|---|
| | **5 min** | **15 min** | |
| Cortisol ($\mu$g/dL) | | | |
| Pull horses | $8.88 \pm 4.78$ [a] | $6.25 \pm 2.72$ [b] | 0.0463 |
| Helper horses | $10.23 \pm 5.18$ [a] | $7.43 \pm 2.16$ [b] | 0.0349 |
| Total WBC ($\times 10^3/\mu$L) | | | |
| Pull horses | $9.07 \pm 1.54$ | $8.58 \pm 1.73$ | 0.4494 |
| Helper horses | $9.46 \pm 2.57$ | $8.49 \pm 1.49$ | 0.1360 |
| NEU ($\times 10^3/\mu$L) | | | |
| Pull horses | $6.03 \pm 0.90$ | $5.61 \pm 0.92$ | 0.3749 |
| Helper horses | $6.68 \pm 2.12$ | $5.88 \pm 1.00$ | 0.0976 |
| LIN ($\times 10^3/\mu$L) | | | |
| Pull horses | $2.71 \pm 1.32$ | $2.70 \pm 1.11$ | 0.9523 |
| Helper horses | $2.45 \pm 0.59$ | $2.37 \pm 0.59$ | 0.8052 |
| HR (beats/min) | | | |
| Pull horses | $61.7 \pm 39.9$ | $74.5 \pm 47.9$ | 0.1012 |
| Helper horses | $64.0 \pm 29.3$ | $59.4 \pm 30.4$ | 0.5515 |

Note: Data expressed as mean values and standard deviation. Different letters in the same line denote significant differences by *t*-test ($p < 0.05$). WBC, white blood cell count; NEU, neutrophils count; LIN, lymphocytes count; HR, heart rate.

The HRV indexes are described in Table 3, with significant low values in mean RR, rMSSD and TINN for helper horses when running with 15-min interval between races. Polar analysis also revealed an HRpeak of ~202 bpm and an HRmed of ~170.4 bpm in M1 and an HRpeak of ~204 bpm and an HRmed of ~159.8 bpm in M2 for pull horses ($p > 0.05$). For helper horses, we observed an HRpeak of ~188.1 bpm and an HRmed of ~154.6 bpm in M1 and an HRpeak of ~144.8 bpm and an HRmed of ~112.2 bpm in M2 with significant differences between M1 and M2 for both measures ($p < 0.05$).

**Table 3.** Media (median) pull and helper horses analysis of HRV parameters according to rest interval time between races (5 min and 15 min).

| | Pull Horses | | | Helper Horses | | |
|---|---|---|---|---|---|---|
| | **5-min Interval between Races** | **15-min Interval between Races** | **p** | **5-min Interval between Races** | **15-min Interval between Races** | **p** |
| | | | Time domain | | | |
| Mean RR (ms) | 737 (629) | 747 (606) | 0.914 | 809 (707) [a] | 714 (518) [b] | 0.004 |
| SDNN (ms) | 868 (756) | 875 (653) | 0.452 | 1123 (952) | 965 (892) | 0.104 |
| rMSSD (ms) | 984 (837) | 971 (797) | 0.914 | 1256 (1136) [a] | 1040 (1127) [b] | <0.001 |
| NN50 (beats) | 17.5 (16.0) | 16.0 (16.0) | 0.294 | 22.8 (23.0) | 28.6 (28.5) | 0.102 |
| pNN50 (%) | 53.6 (55.6) | 56.3 (65.5) | 0.294 | 57.8 (61.0) | 55.2 (61.0) | 0.438 |
| RRtri | 21.27 (16.0) | 18.92 (16.0) | 0.450 | 26.2 (26.8) | 25.9 (28.0) | 0.674 |
| TINN | 3489.7 (4600.5) | 3492.1 (3543.0) | 0.920 | 4681.0 (4546.0) [a] | 3584.0 (4361.0) [b] | <0.001 |
| SI | 4.63 92.80) | 3.01 (3.01) | 0.434 | 1.63 (1.50) | 2.34 (1.75) | 0.314 |
| | | | Frequency domain | | | |
| VLF (%) | 32.3 (30.2) | 23.8 (15.3) | 0.104 | 21.5 (12.6) | 23.6 (16.8) | 0.442 |
| LF (%) | 54.7 (59.3) | 60.3 (63.9) | 0.104 | 63.0 (65.0) | 62.2 (66.4) | 0.823 |
| HF (%) | 13.0 (15.3) | 15.5 (17.5) | 0.438 | 15.6 (16.6) | 15.3 (16.6) | 0.789 |
| | | | Poincaré plot | | | |
| SD1 | 735 (598) | 716 (570) | 0.778 | 892.3 (701.7) | 798.8 (686.7) | 0.904 |
| SD2 | 901 (800) | 883 (802) | 0.678 | 1100.5 (950.2) | 1094.0 (978) | 0.886 |

Note: Different letters mean differences ($p < 0.05$) between columns by Friedman test. Abbreviations: Mean RR = interval between R wave peaks (two consecutive heart beats); SDNN = standard deviation of all normal R-R intervals of the dataset; rMSSD = the square root of the mean of the sum of the squares differences between adjacent normal R-R intervals; NN50 = consecutive R-R intervals that differed more than 50 ms; pNN50 = percentage of consecutive R-R intervals that differed more than 50 ms; ms = milliseconds; TINN = triangular interpolation of NN interval histogram; VLF = very-low-frequency ranges in the frequency domain analysis; HF = high-frequency ranges in the frequency domain analysis; LF = low-frequency ranges in the frequency domain analysis; SD1 and SD2 are, respectively, the short and long-term Poincare plot indexes.

## 3. Material and Methods

The project was approved by the Ethics Committee for Animal Use at the Federal University of Bahia (CEUA-UFBA), Protocol 08/2018.

### 3.1. Animals

Ten healthy Quarter horses were used with an average age of $8.9 \pm 4.3$ years old (range: 4.5–13 years old) and weighing $441.3 \pm 25.0$ kg. These animals belong to the equine training center located in Camaçari-BA (12°48′48.6″ S 38°16′31.3″ O) and Dom Macedo Costa-BA (12°54′12.9″ S 39°11′40.8″ O). All animals underwent a similar husbandry program. Diet consisted of Transvala (*Digitaria decumbens*) and/or Tifton (*Cynodon dactylon* vr. Dactylon) hay *ad libitum* and commercial concentrate (1.5 kg/100 kg body weight, with 12% crude protein) divided in two or three daily portions, according to competition periods. Inorganic mineral salt mixture was offered twice a week, and water was always available.

All horses were physically active, and in the previous six months, they had been training regularly for *vaquejada* practice. The training program was similar on both properties and consisted of 40 min of walking, alternating with galloping, 3–4 times per week; 4–6 races with bulls, 2–3 times per week; and rest or competitions during the weekends. During the trainings, each helper horse worked with two pull horses, so they worked twice as much as pull horses. Training intensity was usually increased in the weeks before a competition. Individual programs associated with the opening of the gate, the race until the marked lines and at the fall area were also implemented to improve animals' technique.

### 3.2. Exercise

The *vaquejada* simulation tests (VSTs) performed were based on previously described protocols [16] and followed the rules of the *Associação Brasileira de Vaquejada* (Brazilian Association of *Vaquejada*—ABVAQ) [17]. A helper horse/cowboy and a pull horse/cowboy ran three times on a soft sand track (130–150 m) with a bull with the former ones keeping the bull running in line and the latter ones galloping and working to pull the bull down at a 100 m mark. Each pair of horses executed two VSTs (M1 and M2) at least 5 days apart one from another. On the first one (M1), they ran with a 5-min rest between races; on the second (M2), they had a 15-min rest between races. Each bull ran only once. The two physical trials were performed in the morning, between 0700 and 1100, during February and March (Summer season in the southern hemisphere) with local mean temperature of 30 °C and a mean relative humidity of ~72%, which is typical of tropical regions. The sand track was dry, and all animals were ridden by their usual riders, according to the welfare precepts described by Coelho et al. [18].

### 3.3. Heart Rate, Heart Rate Variability and Speed

During the VSTs, all horses used an integrated heart rate (HR) and GPS monitoring system (H10 sensor and M430 frequency meter, Polar Electro, Lake Success, NY, USA) that recorded RR intervals every second (1 s). The recorded data were transferred and analyzed using Polar Flow software 6.18.0 (Polar Electro, Lake Success, NY, USA). The horses' HRV was analyzed with Kubios HRV Standard software 3.5.0 (Biomedical Signal Analysis Group, Kuopio, Finland). From the RR intervals, the following time-domain measures were obtained: mean RR interval, standard deviation (SD) of the RR intervals, root mean square (rMSSD) of the successive differences in the RR intervals, NN50 (number of successive RR interval pairs that differed by more than 50 ms), pNN50 (relative number of successive RR interval pairs that differed by more than 50 ms) and HRV triangular index (the integral of the RR interval histogram divided by the height of the histogram). The frequency-domain analysis was performed using the Fast Fourier Transform method with the sampling frequency set at 8 Hz. The power in the heart rate spectrum was divided into three different frequency bands: very low-frequency power (VLF, 0 to 0.04 Hz), low-frequency power (LF, 0.04 to 0.15 Hz) and high-frequency power (HF, 0.15 to 0.4 Hz). The non-linear properties of the horses' HRVs were studied through a Poincaré plot (SD1 describing the short-term variability, SD2 describing the long-term variability, and SD2/SD1 ratio) and approximate entropy (ApEn), which provided a measure of the irregularity of the signal. Detrended fluctuation analysis (DFA) was performed to determine the long-range correlations in the non-stationary physiological time series, yielding both short-term fluctuation ($\alpha$1) and long-term fluctuation ($\alpha$2) slopes.

### 3.4. Clinical and Blood Analysis

On both days (M1, M2), HR (heart rate) was recorded at T0 (before exercise, at rest, inside the box), T1 (immediately after the three races), and T2 (at 240 min of recovery, at the box). HR was also measured in all animals at 30 min of recovery.

Blood samples were aseptically obtained from the jugular vein using disposable hypodermic needles (25 mm × 0.8 mm) and 4 mL tubes containing K3-EDTA for determining the packed cell volume (PCV) and white blood cell total count (WBC), neutrophils and

lymphocytes counts; and 9 mL tubes without anticoagulant were used for cortisol measurement. Immediately after collection and while still in the training centers, serum was obtained after centrifugation for 15 min at 2 g (80-2B centrifuge, Daiki, São Paulo, Brazil). K3-EDTA tubes were transported to the Veterinary Clinical Laboratory using a cooler with ice for immediate processing by a hematological analyzer (BC 2800, Mindray Biomedical Electronics, Shenzhen, China) and examination of blood smears with a mean interval of 4 h between blood collection and laboratory analysis. Serum samples were sent to Biopa (UFRPE, Recife, Brazil) for cortisol determination through ELISA (Monobind Inc., Lake Forest, IL, USA), with sensitivity of 95%, curve range between 0.4 and 95 µg/dL, intra-assay coefficient of variation of 6.4% and inter-assay coefficient of variation of 7.0%.

*3.5. Statistical Analysis*

Results were analyzed using the SAS® OnDemand for Academics program (SAS 9.1, SAS Institute Inc., Cary, NC, USA). All data were evaluated for normality using a Kolmogorov–Smirnov test. Analysis of variance (PROC GLM) for repeated measures followed by comparison between means (*t*-test and Tukey test) was performed to evaluate the possible influence of VSTs on both categories of athletes in M1 and M2 and assess differences between pull and helper horses (animal category). Mean general values for M1 and M2 for each parameter were calculated to compare the possible influence of rest intervals between races on stress indexes (M1 vs. M2). Results are expressed as mean and standard deviation, and values of $p < 0.05$ were considered significant. The Pearson correlation test (PROC CORR) was used to evaluate the strengths of association between the leukocytes, neutrophil and lymphocyte counts, HR, and cortisol; strong correlations occurred when the $r^2$ value ranged from 0.7 to 0.99 and $p < 0.05$.

**4. Discussion**

The assertion that the two equine athletes involved in a *vaquejada* event have different physical demands [16] was confirmed by the different responses observed for the stress parameters.

Cortisol is an acceptable parameter to evaluate physiological and physical stress [5,19]. Its release during physical exercise promotes the emotional and physical changes necessary to mobilize energy [9,20,21]. So, a transitory increase in cortisol levels is expected to help animals deal with the challenge of exercise and competitions, promoting substrate mobilization and behavioral modifications [6,22]; however, this is not always observed. Cortisol release depends on the intensity and duration of physical effort, and its concentration decreases after training [5,11]. Thus, lower cortisol levels are generally associated with fitness [5,9,13,20,23] and learning efficiency [7]. The cortisol levels of helper horses in both VSTs and of pull horses in M2 denote a mental and physiological adaptation to the imposed level of physical effort [6,21,23]. Furthermore, lower values of cortisol for both category of horses with a 15-min rest between races confirms a positive stress (eustress) resulting from *vaquejada* practice [13]. To avoid any anticipatory effects of exercise on cortisol basal levels (pre-effort), T0 samples were obtained in the box.

Stressful events, such as *vaquejada* exercise, can lead to leukocytosis and changes in other inflammatory biomarkers [24]. In Thoroughbred horses, an increased WBC during exercise was related to cortisol [25] despite the number of leukocytes remaining unchanged under the influence of short-term loads [26]. No significant differences were observed in the horses' leukograms in response to both VST, which indicate a lower physical and psychological stress level [9] as well as a good fitness level of all horses. This is also supported by the cortisol measurements obtained.

Time domain measures of HRV, especially the root mean square of successive differences (rMSSD) between the consecutive interbeat intervals, have been associated with parasympathetic nervous system (PNS) activity in horses [20,27,28]. This greater parasympathetic action can be considered a performance marker [29], since vagal activation allows the heart to be more responsive and sensitive to changing environmental demands, such as physical exercise [28,30]. In the present research, significantly lower values of rMSSD

and mean R-R for helper horses were recorded in M2. These findings are associated with a greater sympathetic action when the three races, which makes a *vaquejada* cycle, were performed with a 15-min interval between them. Thereby, it could be thought that this is related to a reduction in the performance of this category of athletes, since some authors suggested that parasympathetic action may be increased because of a training program [13,31].

However, results are contradictory when studying HRV and fitness mainly regarding assessment carried out during the practice of physical activity. An activated PNS dominance has already been described even in untrained horses [32]. Furthermore, most of the studies used horses participating in aerobic exercises, such as dressage [13]. For horses that practice strength and power disciplines (anaerobic efforts/high intensity and short duration), such as *vaquejada*, this is not yet clear, like what happens for human athletes in similar sports categories [33]. Furthermore, high-performance human athletes showed an HRV reduction not associated with fatigue; indeed, reduced HRV even with effective training was explained by parasympathetic saturation [28], which was a phenomenon not yet studied in horses in which a sustained parasympathetic control of the sinus node eliminated respiratory heart modulation, reducing HRV [28].

Another possible explanation for reduction in some HRV parameters in helper horses would be the effect of stress [21,34]. The reduction in rMSSD and R-R interval was described in response to a training program for young horses with lower values associated with the presence of the rider [23], in which animals were facing something new (greater physical demand and acceptance of the rider's presence) [34]. To clarify this, an analysis of stress markers such as cortisol and leukogram was carried out. As previously discussed, such an analysis was compatible with organic adaptation to the imposed effort.

The reduction in HRV parameters (rMSSD and mean R-R) observed for helper horses could be associated with the physical effort load. High-intensity interval training (HIIT) using a treadmill has been associated with HRV reduction in racehorses [30,35] and trotting horses [36]. Also, horses working in a water treadmill under different loads and water levels showed a reduction in HRV parameters indicating a sympathetic domain [37]. At intense effort (HR > 120–130 bpm), such as *vaquejada* exercise (HRmax~144.8 bpm), other non-neural mechanisms may become more important than autonomic modulation in influencing HR and HRV [35]. HRV may be influenced by respiratory gait entrainment, energetics of locomotion and the work of breathing during high-velocity repetitions [37,38], so its analysis in the frequency domain would appear to be of potential value as a non-invasive means of assessing the autonomic modulation of HR only at low-intensity exercises [37]. However, studies involving power equine sports, like *vaquejada*, during physical effort itself and at recovery time, must be conducted for an adequate comprehension and better use of this practical tool on performance and welfare evaluation.

The considerable inter-horse variation and large individual differences have already been described when studying the relationship between HRV and fitness [30,35], and this may explain the differences observed between pull and helper horses. Previous studies already indicate that there are two different athletes used in the same equine discipline [16,39]. Understanding this is crucial to ensuring the good performance of these animals associated with animal welfare practices.

In a detailed analysis, it is observed that the helper horses significantly increased their HR by 138% in M2, while in M1, this increase was 108.8%. As HR is directly related to exercise intensity [40], such findings confirm the greater intensity practiced by helper horses in M2. Just like human athletes, the intensification of physical effort is only possible when horses have a good physical condition. The good physical condition in helper horses was reflected through a significant reduction in HRpeak and HRmed recorded during exercise, denoting a possible greater cardiac efficiency [16,28]. Beyond this, all horses used in the present research recovered their pre-exertion HR values already with 30 min of recovery with both rest intervals tested between races, which can denote a good athletic conditioning of horses used in the present experimental protocol [39] while also respecting the premises

of welfare [18]. In pull horses, the intensity of physical activity was higher, which was reflected in HR, with HRpeak > 200 bpm, and justified by the greater force required to pull the bull down [12,16]. However, HRV indices remained constant regardless of the interval period between races.

Optimal physical condition includes not only a sequence of training exercises but also an adequate period of rest between training sessions and between/during competitions [41,42]. The physical demand of helper horses is due to their greater use in competitions, as, generally, for each pull horse used, three pulling horses are worked [39]. Our previous research with the same horses demonstrated that such athletes must be trained differently and, furthermore, that under these conditions, helper horses show benefits when they have a 15-min rest period between races [16]. The present work corroborates these findings.

## 5. Conclusions

The results showed different responses between the two categories of equine athletes studied. From the cortisol data, it can be suggested that the 15-min rest interval between the races allowed a more complete recovery than the 5-min rest interval for both categories of equine athletes used in *vaquejada*. However, the lower HRV indices and the more intense increase in HR in M2 reinforce the relevance of this statement for helper horses. The longer rest time allowed the organic recovery necessary for these animals to impose a greater applied physical effort load, with less cardiac effort, during VST, which is a fact that guarantees good performance and welfare.

Further studies are necessary for a better use of stress predictors in horses used for high-intensity equine disciplines.

**Author Contributions:** T.R.P.S. and L.N.S. contributed to the study design and execution and data collection. C.S.C. contributed to the study design, data interpretation, manuscript preparation and supervision. T.M.S., J.S., V.R.C.S., R.F.S. and H.C.M.F. contributed to data interpretation and manuscript preparation. All authors have read and agreed to the published version of the manuscript.

**Funding:** This research received no external funding.

**Data Availability Statement:** Data are unavailable due to privacy or ethical restrictions.

**Conflicts of Interest:** The authors declare no conflict of interest.

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
