# Peer review of "Analysis of Stress Predictors in Vaquejada Horses Running with Different Interval Rest Periods"

_stresses, doi:10.3390/stresses3040058_

Round 1
Reviewer 1 Report
Comments and Suggestions for Authors
Interesting article on the stress responses of horses used for Vaquejada. Not being familiar with this sport, I had to find a few videos to view, although the authors did an adequate job of describing it in the introduction. The authors used adequate and varied stress parameters on the horses. Was there a reason the authors used serum for cortisol rather than salivary cortisol? Were all the horses acclimated to jugular venipuncture prior to the study? There is the potential for a stress response associated with jugular venipuncture. Anticipatory elevations in cortisol have been documented. Was there any observations on horse behaviors prior to the VSTs? This article would be more complete if the authors would have added an additional study on the stresses of the bulls being used, even if it was as simple as pre and post fecal cortisol concentrations. There are a few published, peer review articles that have looked at the adaptation and stresses on roping steers / calves that have indicated that these animals adapt quickly to the activity and once ‘learned’, do not appear to undergo stress responses. This article is one of only a few studies that have concentrated on studying stress in horses performing anaerobic exercise, although the sport selected is fairly unique.

Comments on the Quality of English LanguageOverall the article was fairly easy to understand although improvements could be made per suggestions included in the uploaded file. Some spacings before and after numerical data should be standardized per journal format.
Author Response
Comments and Suggestions for Authors – REVIEWER 1
Analysis of Stress Predictors in Vaquejada Horses Running with Different Interval Rest Periods
Interesting article on the stress responses of horses used for Vaquejada. Not being familiar with this sport, I had to find a few videos to view, although the authors did an adequate job of describing it in the introduction. The authors used adequate and varied stress parameters on the horses.
Thank you for your considerations.
Was there a reason the authors used serum for cortisol rather than salivary cortisol? Were all the horses acclimated to jugular venipuncture prior to the study? There is the potential for a stress response associated with jugular venipuncture. Anticipatory elevations in cortisol have been documented.
All horses were used to venipuncture procedures being easier to get those samples than to try the salivary cortisol. Head handling of such animals is quite complicated, so we prefer the venipuncture.
Was there any observations on horse behaviors prior to the VSTs?
All Quarter Horses used were very calm and very used to procedures.
This article would be more complete if the authors would have added an additional study on the stresses of the bulls being used, even if it was as simple as pre and post fecal cortisol concentrations. There are a few published, peer review articles that have looked at the adaptation and stresses on roping steers / calves that have indicated that these animals adapt quickly to the activity and once ‘learned’, do not appear to undergo stress responses.
Another research group of UFBA made some evaluations however the data is still being processed. But sample collection is difficult.
This article is one of only a few studies that have concentrated on studying stress in horses performing anaerobic exercise, although the sport selected is fairly unique. Thank you again for your considerations.
Specific Comments
Line 27 It is customary to have a space before and after ± and when citing P values (P < 0.05)
Modifications were made and throughout the text.
Line 29 Change significant to significantly or reword sentence, eliminating ...other important findings and just state that rMSSD and R-R were lower....
Modifications were made.
Line 34 Delete 1 before cortisol (in key words)
Modifications were made.
Line 44 Reword sentence....and potentially a decrease in quality of life.
Modifications were made.
Line 61 Delete ‘the’; change its’ to its (possessive pronoun referring to the bull Modifications were made.
Line 68 Capitalize The
Modifications were made.
Line 76 Change weighting to weighing
Modifications were made.
Line 75 Suggested change: Ten healthy Quarter Horses, and eliminate the last clause in sentence.
Modifications were made.
Line 78 Since the authors discuss nutrition and exercise program, why not simply state ‘ Horses were maintained on similar husbandry programs. Then go on to describe nutrition and exercise.
Modifications were made.
Line 90 Change, each animal’s, or animals’ (i.e. grammar error)
Modifications were made.
Line 96 Change last to latter
Modifications were made.
Line 92-99 The language is a bit confusing. Authors state the horses ran three VSTs and then describe M1 and M2. I interpreted the language to mean that each pair of horses ran 3 times during M1 and M2 which were conducted a minimum of 5 days apart. Please clarify.
Modifications were made.
Line 100 Authors could use military time 0700 and 1100 hour rather than stating a.m. Modifications were made.
Line 101 Authors either need to spell out degrees, or use the appropriate symbol (0) which is located in Symbols in WORD
Modifications were made.
Lines 142 T0 and T1 make sense. Does the 30 minutes after recovery refer to after T2 (240 minutes recovery) or does it take place 30 minutes post race (T1)? Please clarify. It was an extra analysis to evaluate heart rate recovery. So, we had T0 (before), T1 (immediately after), 30 min (only for HR) and T2 (240 minutes of recovery).
Line 162 Change was to were (values are plural)
Modifications were made.
Line 145/178-180 Authors refer to total WBC here, use the same terminology in materials and methods. Since neutrophils and lymphocytes are WBC a positive correlation is expected.
Modifications were made.
Lines 230-242 Authors should discuss their cortisol values relative to those cited in the literature. The cortisol ELISA our lab generally uses provides higher cortisol levels compared to those you saw. You may need to elaborate. Although statistically different, do your cortisol levels indicate stress?
Increases after exercise can be considered a stress response to prepare for the effort. Comparisons were made with basal levels for each horse and as we observed an increase, we characterized as a cortisol release due exercise.
Line 255 On the present research, significative lower values of rMSSD 224 and mean R-R for helper horses were recorded at M2. Should this read ‘In the present research, significantly lower values...?
Modifications were made.
Line 266 Add the ‘anaerobic’ to your exercise description since you stated that most other studies have used aerobic exercise.
Modifications were made.
General discussion Somewhere in your discussion you may want to discuss changes in stress parameters in anticipation of a stressor / or event since T0 took place ‘at rest, but in the box’.
Modifications were made.
Line 311 The authors may want to re-examine their conclusion that the horses were fit based on HR data, as HR drops very quickly following intense exercise and will return to normal resting rate within a small window (under 30 minutes). Although the horses used in this study were probably fit and well conditioned, concluding this based on return to resting HR may be somewhat misleading to readers.
Modifications were made.
Line 328 Conclusion is taking liberties stating that the 15 minute rest is ideal as the authors only tested two different rest protocols. Certainly they can state that a 15 minute rest was better than 5 minutes, I’m not sure they can say that it is ‘ideal’. Modifications were made.
Line 349 Delete – in impact, add 33(3):155-160 to Peeters reference
Modifications were made.
Line 443 Where is reference 43 cited in the text??
Modifications were made. Error on format.
Reviewer 2 Report
Comments and Suggestions for Authors
Dear Authors,
The manuscript describes the analysis of stress response in horses to the specific stress stimulus which is vaquejada simulating tests (VST). To the best of my knowledge, this is the first research investigating the effect of VST on horses' welfare using heart rate-related indicators. In my opinion, this manuscript makes a significant contribution to the development of veterinary and animal sciences disciplines and raises the important topic of assessing the welfare of horses during specific, directed exercise other than the classic exercise of racing, sports, school, or leisure horses. The raised issue deserves appreciation. I do not find any serious flaws in this manuscript, and my minor comments are editorial in nature.
The introduction, although short, introduces the reader well to the topic. It reads clearly and is not overloaded.
However, the aim and hypothesis need rewriting. L 62-63 should be reworded as a hypothesis, started with a capital letter, and raised as the first. Then the hypothesis should be followed by the reworded aim of the study. In my opinion, you aim to evaluate the changes in stress indicator values (cortisol level, leukogram, heart rate, and heart rate variability) concerning the vaquejada simulation tests that Quarter horse athletes were subjected to.
Therefore, the conclusions should be slightly reworded to make them more consistent with the objective. Be careful to avoid definitive judgments. I'm afraid that your experience diagram does not "prove the ideal indicator" but only indicates differences in feature values. Be more careful with your conclusions.
Specific comments
L 68 Subsection 2.1 also lacks information on whether the horses were healthy at the start of the study and how they were classified. Typically, proper reference to the veterinary prequalification with citations to the clinical examination protocol is satisfactory. Specific classification criteria are often used for this purpose, the number of horses included and the number of horses excluded.
L 70 The description of the experience is concise but precise. The age of the horses should be supported by a range and not just shown as an average.
L 127 Since the concentration of cortisol in the blood varies in horses during the day, it is necessary to specify the time (range of hours) of sampling.
L 137 For the ELISA tests more details should be provided such as sensitivity, curve range, intra-assay coefficient of variation, and inter-assay coefficient of variation.
L 163 and L 172 The tables are clear and well constructed. The statistical analysis used is correct and properly marked in the tables. My comment here concerns the units in which you present the serum cortisol concentration. The reference unit is nmol/L and cortisol concentration should be presented in such a unit. Consider converting the presented values.
​The results section is presented clearly and in a very well-organized way.
The discussion also requires to be restructured. You should attempt structuring the discussion following a "common standard format" that usually consists of the following points:
a. One sentence summary that highlights the most relevant results.
b. A thorough discussion of each result obtained concerning the corresponding study objective: was the tested hypothesis confirmed or not? Why? What previous evidence supports the specific result or not? It is critical to compare/contrast the result obtained with previous literature in the equine species first and then in veterinary medicine (if not enough data are available for comparison in equine research).
c. Statement of study limitations
d. Future directions
I have the pleasure of assessing your work very positively and requesting acceptance for publication after addressing the minor comments mentioned above.
Author Response
Comments and Suggestions for Authors – REVIEWER 2
Analysis of Stress Predictors in Vaquejada Horses Running with Different Interval Rest Periods
The manuscript describes the analysis of stress response in horses to the specific stress stimulus which is vaquejada simulating tests (VST). To the best of my knowledge, this is the first research investigating the effect of VST on horses' welfare using heart rate-related indicators. In my opinion, this manuscript makes a significant contribution to the development of veterinary and animal sciences disciplines and raises the important topic of assessing the welfare of horses during specific, directed exercise other than the classic exercise of racing, sports, school, or leisure horses. The raised issue deserves appreciation. I do not find any serious flaws in this manuscript, and my minor comments are editorial in nature.
Thank you for your considerations.
The introduction, although short, introduces the reader well to the topic. It reads clearly and is not overloaded.
Thank you.
However, the aim and hypothesis need rewriting. L 66-68 should be reworded as a hypothesis, started with a capital letter, and raised as the first. Then the hypothesis should be followed by the reworded aim of the study. In my opinion, you aim to evaluate the changes in stress indicator values (cortisol level, leukogram, heart rate, and heart rate variability) concerning the vaquejada simulation tests that Quarter horse athletes were subjected to.
Modifications were made.
Therefore, the conclusions should be slightly reworded to make them more consistent with the objective. Be careful to avoid definitive judgments. I'm afraid that your experience diagram does not "prove the ideal indicator" but only indicates differences in feature values. Be more careful with your conclusions.
Modifications were made.
Specific comments
L 74 Subsection 2.1 also lacks information on whether the horses were healthy at the start of the study and how they were classified. Typically, proper reference to the veterinary prequalification with citations to the clinical examination protocol is satisfactory. Specific classification criteria are often used for this purpose, the number of horses included and the number of horses excluded.
We only examined 10 horses. At the first evaluation all proved to be healthy in clinical exams. Usually, exams include a complete physical evaluation and lameness evaluation. Adjustments were made according to both reviewers requests.
L 76 The description of the experience is concise but precise. The age of the horses should be supported by a range and not just shown as an average.
Modifications were made.
L 127 Since the concentration of cortisol in the blood varies in horses during the day, it is necessary to specify the time (range of hours) of sampling.
It is described in line 101. All sample were obtained between 0700-1100.
L 160-162 For the ELISA tests more details should be provided such as sensitivity, curve range, intra-assay coefficient of variation, and inter-assay coefficient of variation.
Data included.
The tables are clear and well constructed. The statistical analysis used is correct and properly marked in the tables. My comment here concerns the units in which you present the serum cortisol concentration. The reference unit is nmol/L and cortisol concentration should be presented in such a unit. Consider converting the presented values.
As cortisol can be expressed in both units we would like to keep as µg/dL, as it was also recommended by the used kit. Also, we think the table and information would be clearly this way (with a smaller number).
The results section is presented clearly and in a very well-organized way.
Thank you.
The discussion also requires to be restructured. You should attempt structuring the discussion following a "common standard format" that usually consists of the following points:
- One sentence summary that highlights the most relevant results.
- A thorough discussion of each result obtained concerning the corresponding study objective: was the tested hypothesis confirmed or not? Why? What previous evidence supports the specific result or not? It is critical to compare/contrast the result obtained with previous literature in the equine species first and then in veterinary medicine (if not enough data are available for comparison in equine research).
- Statement of study limitations
- Future directions
We prepared the discussion considering all the above topics. We began confirming our hypothesis highlighting the different responses observed saving the conclusion for later. Then we discussed each variable linking the observed findings and comparing with previous studies involving not only vaquejada but also other equestrian disciplines. We have made some improvements on the text according to both reviewers’ comments. Hope this works out.
I have the pleasure of assessing your work very positively and requesting acceptance for publication after addressing the minor comments mentioned above.
Thank you again for your important considerations.
Round 2
Reviewer 1 Report
Comments and Suggestions for Authors
Thank you for making the suggested changes in the manuscript. I will be interested to read about the stress data on the bulls at a future time. The authors only tested 2 rest periods so their conclusion is that the 15 minute rest period allowed a more complete recovery compared to the 5 minute rest period. However they should not state that is 'ideal' since they only tested 2 different treatments. Please restate this in the abstract and the conclusion.
Author Response
Comments and Suggestions for Authors – REVIEWER 2 Round 2
Analysis of Stress Predictors in Vaquejada Horses Running with Different Interval Rest Periods
Thank you for making the suggested changes in the manuscript. I will be interested to read about the stress data on the bulls at a future time. The authors only tested 2 rest periods so their conclusion is that the 15 minute rest period allowed a more complete recovery compared to the 5 minute rest period. However they should not state that is 'ideal' since they only tested 2 different treatments. Please restate this in the abstract and the conclusion.
Thank you for your considerations. Modifications were made in lines 31 (Abstract) and 341 (Conclusions).